# DELTA-TRIPLANE TRANSFORMERS AS OCCUPANCY WORLD MODELS

## ABSTRACT

Occupancy World Models (OWMs) aim to predict future scenes via 3D voxelized representations of the environment to support intelligent motion planning. Existing approaches typically generate full future occupancy states from VAE-style latent encodings. In contrast, we propose *Delta-Triplane Transformers* (DTT), a novel 4D OWM for autonomous driving. DTT adopts *temporal triplane* as the occupancy representation, and focuses on modeling *changes* in occupancy rather than dealing with full states. The core insight is that changes in the compact 3D latent space are naturally sparser and easier to model, enabling higher accuracy with a lighter-weight architecture. We first pretrain a triplane representation model that encodes 3D occupancy compactly, and then extract multi-scale motion features from historical data and iteratively predict future triplane deltas. These deltas are combined with past states to decode future occupancy and ego-motion trajectories. Extensive experiments show that DTT achieves a state-of-the-art mean IoU of 30.85, reduces mean absolute planning error to 1.0 meter, and runs in real time at 26 FPS on an RTX 4090. Demo videos and code are provided in the supplementary material.

## 1 INTRODUCTION

World models (Ha & Schmidhuber, 2018b;a) aim to represent the environment, predict future scenes and enable agents to perform advanced motion planning. Recently, 3D occupancy technique (Tong et al., 2023; Tian et al., 2023) has emerged as a structured and spatially rich representation of the environment, enabling constructing finer correlations between occupied voxels and planning decisions. In this paper, we focus on the design of Occupancy World Models (OWMs) for autonomous driving.

In designing OWMs, there are two tasks need to be solved: (i) learning a compact latent representation of raw occupancy data, and (ii) using this representation for scene forecasting and motion planning.

For the first task, most existing OWMs (Wei et al., 2024; Xu et al., 2025b; Zheng et al., 2024a; Yan et al., 2025) mainly use VQ-VAE based BEV representations (Van Den Oord et al., 2017). While VAE has shown success in several generative tasks (Jiang et al., 2023b; Zhang et al., 2024a), its use for occupancy compression requires the latent space of 3D structures to be converted into a 2D BEV plane, which we denote by $xy$. In such a way, objects that appear similar in BEV but differ vertically may become indistinguishable, forcing the BEV-based methods to rely on fine-grained semantic cues. This usually means using more feature channels and thus increased model size. Recent advances have shown that the triplane representation provides an effective alternative for 3D occupancy prediction and scene generation (Khatib

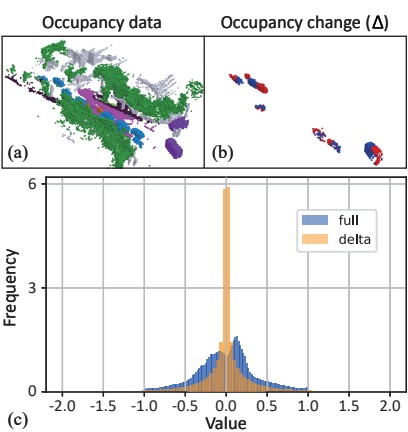

Figure 1: (a-b) Comparison between the full occupancy and occupancy changes (Δ) of a scene. In the change map, red indicates newly appeared voxels, while blue denotes disappeared voxels. (c) Distribution of latent-space feature values for full occupancy versus occupancy changes.

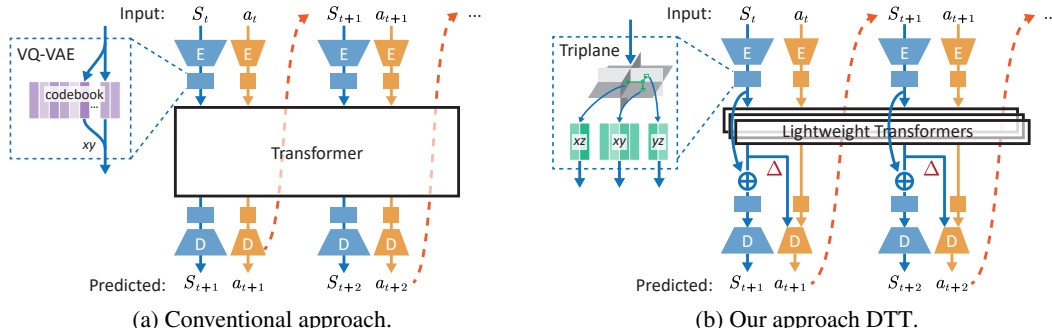

(a) Conventional approach.  (b) Our approach DTT.

Figure 2: Comparison of occupancy world model architectures: (a) Conventional approaches use a pre-trained VAE to compress occupancy state $S$ into a compact BEV representation, which is then combined with previous actions $a$ to predict future BEVs and actions via a large Transformer. (b) Our method, DTT, adopts more compact and precise triplane representations and uses lightweight Transformers to separately predict the triplane changes of each plane. These deltas are then used to for occupancy forecasting and motion planning.

& Giryes, 2024; Hu et al., 2023a; Lee et al., 2024). By incorporating additional $xz$ and $yz$ planes, triplane preserves vertical information. Motivated by this benefit, we propose to adopt the triplane representation for 4D OWMs.

For the second task, the key is how to fully exploit triplanes for 4D occupancy prediction and motion planning. Existing OWMs typically use a large Transformer to model the full occupancy state (Xu et al., 2025b; Wei et al., 2024). In particular, its multi-head attention mechanism is expected to capture the diverse motion patterns of multi-scale objects – for example, the abrupt movements of pedestrians versus the more inertial dynamics of large vehicles. This increases model complexity and parameter count, and makes long-horizon predictions more susceptible to error accumulation (as confirmed in our experiments). In contrast, we observe that occupancy changes are inherently sparse and thus easier to model. Figure 1a depicts this by comparing the full occupancy of a scene with the corresponding occupancy changes across two adjacent frames, showing how key elements of interest are more distinct in the change map. Figure 1b further contrasts their distributions in the latent space, where the $x$ axis is a scalar feature value. While full occupancy states exhibit a scattered distribution, occupancy changes states are much more tightly concentrated around zero, which reduces variance and simplifies learning.

Building on this insight, we propose *Delta-Triplane Transformers* (DTT), a novel 4D OWM model that predicts future states incrementally rather than in full. This leads to a much lighter-weight architecture that runs faster and achieves higher predictive accuracy. Figure 2 compares the conventional model architecture with our approach. In particular, we extend the triplane representation into the temporal domain and use separate Transformers to predict changes on each plane auto-regressively. These deltas are then used as sparse queries to attentively produce planning outputs. DTT achieves state-of-the-art (SOTA) performance. For example, compared with DOME (Gu et al., 2024), it reduces cumulative errors in long-term occupancy prediction, improving mIoU from 27.10 to 30.85 and IoU from 36.36 to 74.58. For motion planning, it attains the lowest average error of 1.0 meter and the lowest collision rate of 30%. In addition, DTT is efficient, running at 26 FPS on an RTX 4090. In summary, our contributions are three-fold:

• We introduce DTT, a novel 4D autoregressive OWM that forecasts future scenes through incremental changes instead of full occupancy states.

• DTT leverages compact triplane representations and lightweight multi-scale Transformers to predict sparse deltas on each plane, which are fused for occupancy forecasting and motion planning.

• Extensive experiments on the nuScenes (Caesar et al., 2020) and Occ3D (Tian et al., 2023) datasets validate our SOTA performance in terms of occupancy forecasting, motion planning, and real-time execution.

## 2 RELATED WORKS

**3D occupancy reconstruction.** 3D occupancy reconstruction methods (Tong et al., 2023; Tian et al., 2023; Wei et al., 2023; Liu et al., 2024; Marinello et al., 2025) aim to represent the environment as a 3D voxel grid where each voxel encodes both geometric and semantic information. The field was pioneered by SSCNet (Song et al., 2017) for indoor scenes using depth sensors, and later extended to outdoor camera-based settings by MonoScene (Cao & De Charette, 2022). These approaches primarily utilize multi-camera RGB images (Li et al., 2023a; Huang et al., 2023) or LiDAR point clouds (Cao et al., 2024; Xia et al., 2023) to infer voxel-wise occupancy and semantics in an agent-centric coordinate system. A core challenge lies in building accurate correlations between raw sensor data and 3D voxel space. Once an accurate occupancy reconstruction is achieved, further learning of the temporal dynamics of scene changes becomes necessary (Li et al., 2024a).

**4D occupancy prediction.** To anticipate future scene changes, various 4D occupancy prediction methods have been proposed to capture the temporal dynamics of scene evolution. Some (Lu et al., 2021; Mersch et al., 2022; Khurana et al., 2023) predict future sensor-level data, which is subsequently voxelized into occupancy results, while others (Ma et al., 2024; Chen et al., 2025; Xu et al., 2024; 2025a) directly forecast occupancy outcomes from historical observations. These methods mainly focus on reducing spatio-temporal biases on future occupancy predictions. However, they overlook the use of predicted scenes for effective and comprehensive motion planning.

**End-to-end autonomous driving.** Conventional end-to-end autonomous driving systems (Hu et al., 2022; 2023b) follow a pipeline of perception, prediction, and planning, which are typically decoupled and optimized separately. The perception module produces structured representations – such as 2D bounding boxes, BEV features, and occupancy grids – that provide accurate ego localization and rich scene semantics. The prediction module infers the intentions of nearby traffic participants and forecasts future scene evolution. The planning module (Li et al., 2024b) then generates safe and feasible trajectories based on this information. Many recent approaches achieve remarkable performance by leveraging more supervisions (Zheng et al., 2024b), high-definition maps (Zheng et al., 2024b; Wen et al., 2024), or richer intention reasoning (Chen et al., 2024; Zheng et al., 2025; Wen et al., 2024). In contrast, our work relies solely on 3D latent triplane changes as a compact conditioning signal for both scene forecasting and motion planning, within the domain of occupancy world models introduced later.

**World models for autonomous driving.** World models (Ha & Schmidhuber, 2018b;a) aim to compress high-dimensional scene representations to capture the temporal dynamics of scene transitions, facilitating both future scene predictions and motion planning for the agent. In autonomous driving, existing models (Jiang et al., 2023a; Zhang et al., 2024b) typically map surrounding traffic participants to the BEV perspective to predict instance-level tracklets or directly use diffusion models (Wu et al., 2024; Wang et al., 2024b;a; Gao et al., 2025; Jia et al., 2023) to generate pixel-level future driving views. These methods derive control signals for the agent from current observations and predicted surroundings, but they rely solely on 2D BEV or image space, which limits the ability to establish fine-grained, efficient correlations between scene changes and motion planning. Recent world models (Wei et al., 2024; Xu et al., 2025b; Zheng et al., 2024a; Yan et al., 2025; Gu et al., 2024) have leveraged 3D occupancy data to address this issue. However, they typically use VAE-series (Van Den Oord et al., 2017; Kingma et al., 2013) models for environment compression, which often neglect original 3D geometric information and compromises reconstruction accuracy. Additionally, they rely on Transformers (Vaswani et al., 2017) to forecast the entire future scene instead of incremental changes, leading to significant error accumulation.

## 3 METHODOLOGY

### 3.1 FORMULATION

Next, we provide the formulation of Occupancy World Model (OWM) (Zheng et al., 2024a). OWM primarily receives a sequence of scene representations and motion actions from past $\tau_p$ frames up to the current timestep $t$, such that $S^t \in \mathbb{R}^{H \times W \times L}$ represents the occupancy data of the agent-centric surrounding environment, with $H$, $W$, and $L$ denoting the height, width, and length, respectively, and $a^t \in \mathbb{R}^2$ denotes a transition-related motion command. The goal of OWM is to establish a

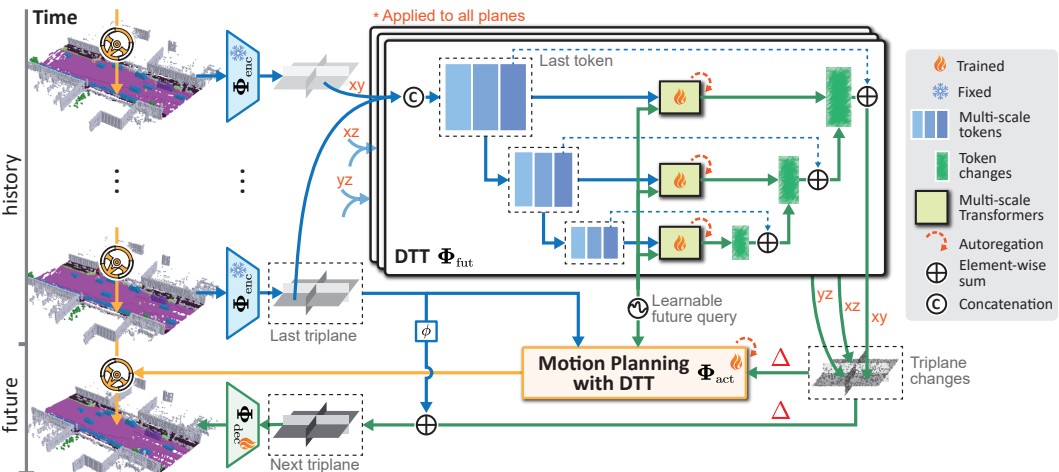

Figure 3: Workflow of DTT. DTT first pre-trains compact triplane representations of occupancy data. It then applies multi-scale Transformers to model temporal dynamics within each plane and predict future triplane changes ($\Delta$). Finally, these changes, combined with the previous triplane, are used to generate future occupancy results and motion proposals.

stochastic mapping, $\mathbf{\Phi}$, that associates past occupancy data and actions with future $\tau_f$ frames of occupancy data and action proposals. Formally:

$$\hat{S}^{t+1:t+\tau_f}, \hat{a}^{t+1:t+\tau_f} = \mathbf{\Phi}(S^{t-\tau_p:t}, a^{t-\tau_p:t}). \tag{1}$$

To achieve this, we first pretrain a compact triplane representation of the raw occupancy data with an auto-encoder. The encoder and decoder, parameterized by $\mathbf{\Phi}_{\text{enc}}$ and $\mathbf{\Phi}_{\text{dec}}$, are defined as:

$$\hat{s}^t = \mathbf{\Phi}_{\text{enc}}(S^t), \quad \hat{S}^t = \mathbf{\Phi}_{\text{dec}}(\hat{s}^t), \tag{2}$$

where $s^t = [s^t_{xy}, s^t_{xz}, s^t_{yz}]$ contains three orthogonal feature planes. Specifically, $s^t_{xy} \in \mathbb{R}^{c \times w \times l}$, $s^t_{xz} \in \mathbb{R}^{c \times h \times w}$, and $s^t_{yz} \in \mathbb{R}^{c \times h \times l}$, with $c$ the channel dimension and $(h, w, l)$ the spatial resolutions along the three axes. $\hat{S}^t$ is the reconstructed occupancy data.

Once we obtain the latent scene representations $s^t$, we can predict future latent states with incremental changes $\Delta \hat{s}^{t+1}$ produced by the scene forecasting model $\mathbf{\Phi}_{\text{fut}}$. These are combined with the previous latent state $s^t$ and decoded back to future occupancy outcome $\hat{S}^{t+1}$ using $\mathbf{\Phi}_{\text{dec}}$. Since $\Delta \hat{s}^{t+1}$ encodes changes between consecutive latent states, it can serve as sparse queries for the planning model $\mathbf{\Phi}_{\text{act}}$ to generate future actions $\hat{a}^{t+1}$.

Note that our OWM, $\mathbf{\Phi} = \{\mathbf{\Phi}_{\text{enc}}, \mathbf{\Phi}_{\text{dec}}, \mathbf{\Phi}_{\text{fut}}, \mathbf{\Phi}_{\text{act}}\}$, operates in an autoregressive fashion, iteratively using previously predicted outcomes as part of the historical data to forecast future scenes and provide motion proposals. Figure 3 illustrates our design, detailed below.

## 3.2 DTT AS OWM

### 3.2.1 PRE-TRAINING TRIPLANE REPRESENTATIONS FOR OWM

To enable faster and more accurate predictions of future scenes $S^{t+1:t+\tau_f}$, OWM requires a compact yet accurate representation of the high-dimensional voxelized occupancy scene. To achieve this, we employ the triplane technique (Lee et al., 2024), which is widely used in volume rendering (Shue et al., 2023; Hu et al., 2023a; Song et al., 2024; Huang et al., 2023), to compress raw occupancy data in an orthogonal decomposition fashion.

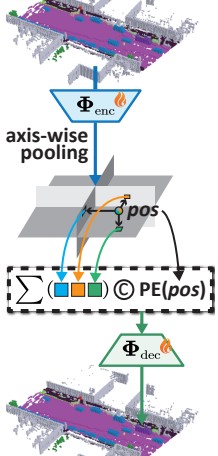

Figure 4: Pretrained triplane for OWM.

Specifically, as shown in Figure 4, the occupancy data $S^t$ is first encoded by $\mathbf{\Phi}_{\text{enc}}$ into $s^t$. Then, an axis-wise average pooling operation is applied to obtain three orthogonal feature planes, $s^t =$

$[s_{xy}^t, s_{xz}^t, s_{yz}^t]$, where $s_{xy}^t \in \mathbb{R}^{c \times w \times l}$, $s_{xz}^t \in \mathbb{R}^{c \times h \times w}$, and $s_{yz}^t \in \mathbb{R}^{c \times h \times l}$. To decode the occupancy data, each 3D point $pos = (x, y, z)$ queries its corresponding features from the three planes. These features are summed, concatenated with the positional encoding $\mathrm{PE}(pos)$, and passed through $\boldsymbol{\Phi}_{\mathrm{dec}}$ to predict the semantic label. This process is formalized in Eq. 2, and detailed network structures are provided in the supplementary material.

We pretrain $\boldsymbol{\Phi}_{\mathrm{enc}}$ and $\boldsymbol{\Phi}_{\mathrm{dec}}$ to obtain compact yet high-fidelity latent triplane representations using:

$$J_{\mathrm{enc,dec}} = \mathbb{E}_{t \sim \mathcal{T}, pos \sim S^t}[\mathcal{L}_{occ}(\boldsymbol{\Phi}_{\mathrm{dec}}(\boldsymbol{\Phi}_{\mathrm{enc}}(S^t)), S^t)], \tag{3}$$

where $\mathcal{T}$ represents the collection of all timesteps in the occupancy dataset, and $\mathcal{L}_{occ} = \mathcal{L}_{ce} + \lambda \mathcal{L}_{lz}$. Here, $\mathcal{L}_{ce}$ and $\mathcal{L}_{lz}$ denote the cross-entropy and Lovasz-softmax losses (Berman et al., 2018; Lee et al., 2024), receptively, and $\lambda$ is the trade-off factor.

The triplane representation, compared to the tokens generated by the VQ-VAE in OccWorld (Zheng et al., 2024a) and the MS-VAE in OccLLM (Xu et al., 2025b), retains 3D structural information while achieving a more compact latent space (see Table 3 for evaluation). This not only makes our OWM more lightweight but also reduces the cumulative prediction error over time, as detailed later.

### 3.2.2 DELTA-TRIPLANE TRANSFORMERS (DTT)

DTT follows the autoregressive prediction paradigm similar to GPT-like models but differs by leveraging historical triplanes to predict future triplane changes rather than full state predictions. These changes are then decoded into future scenes and motion trajectories. At the timestep $k \in \{1, \cdots, \tau_f\}$ for autoregressive forecasting, given $\tau_p$ historical triplane frames $s^{k-\tau_p:k}$, the objective of DTT is to capture the complete temporal dynamics of the scene, particularly adapting to object motions at different scales. To this end, DTT performs two steps: (i) predicting plane-specific future changes $\{\Delta \hat{s}_i^k \mid i \in \{xy, xz, yz\}\}$ with multi-scale Transformers, and (ii) aggregating these changes with the previous state and aligning the three planes through a fine-tuned decoder $\boldsymbol{\Phi}_{\mathrm{dec}}$.

In particular, we design the future prediction module as three plane-specific models: $\boldsymbol{\Phi}_{\mathrm{fut}} = \{\boldsymbol{\Phi}_{\mathrm{fut}_{xy}}, \boldsymbol{\Phi}_{\mathrm{fut}_{xz}}, \boldsymbol{\Phi}_{\mathrm{fut}_{yz}}\}$. Each predictor $\boldsymbol{\Phi}_{\mathrm{fut}_i}$ is implemented with Transformers (Vaswani et al., 2017) operating at multiple scales. The predictors share the same architecture but use separate learnable parameters and different input sizes, depending on the plane and scale, as illustrated in Figure 3. We denote by $k \in \{1, \cdots, \tau_f\}$ the timestep for autoregressive forecasting. Take the $xy$-plane predictor as an example: the input $s_{xy}^k$ is first downsampled using a UNet (Ronneberger et al., 2015)-style encoder, producing $V$ scales of features. At scale $v \in V$, the feature is denoted as $s_{xy}^{k,v} \in \mathbb{R}^{c_{xy}^v \times w_{xy}^v \times h_{xy}^v}$, which is flattened into $w_{xy}^v \times h_{xy}^v$ tokens and passed to a Transformer encoder to build spatio-temporal memory. A learnable future query $Q_{xy}^k$, with the same dimension as $s_{xy}^{k,v}$, is then used in the Transformer decoder for cross-attention, generating token changes from the current step to the next. Finally, token changes across all scales are fused by UNet-style upsampling to produce the plane's feature change $\Delta \hat{s}_{xy}^{k+1}$, which is combined with the previous state $\hat{s}_{xy}^k$ via a $1 \times 1$ convolution $\phi$. The same procedure applies to the other planes. Detailed architecture and hyperparameters are provided in the supplementary material.

The overall forecasting process at autoregressive timestep $k$ is defined as:

$$\Delta \hat{s}_i^k = \boldsymbol{\Phi}_{\mathrm{fut}_i}(\hat{s}_i^{k-\tau_p:k}, Q_{s_i}^k), \tag{4}$$

$$\hat{s}_i^k = \Delta \hat{s}_i^k + \phi(\hat{s}_i^{k-1}), \tag{5}$$

$$\hat{S}^k = \boldsymbol{\Phi}_{\mathrm{dec}}(\hat{s} := \{\hat{s}_i^k\}), \tag{6}$$

where $i \in \{xy, xz, yz\}$ and $Q_i^k$ is the learnable query at timestep $k$ in the $i$-th plane.

Since the triplane occupancy representations are pretrained, the ground-truth (GT) future triplanes can be directly used to supervise DTT. In this setting, we freeze the encoder and fine-tune only the decoder $\boldsymbol{\Phi}_{\mathrm{dec}}$. This ensures a stable and geometry-consistent triplane initialization, while allowing the decoder to adapt specifically to future dynamics. Such decoupling also mitigates representation drift and reduces misalignment across independently predicted planes, leading to more coherent multi-plane aggregation.

Since the triplane occupancy representations are pretrained, the ground-truth (GT) future triplanes can be directly used to supervise DTT. In this setup, we freeze the encoder and fine-tune only the

decoder $\Phi_{\text{dec}}$. This provides a stable and geometry-consistent triplane initialization, while allowing the decoder to adapt specifically to future dynamics. Such decoupling also mitigates representation drift and reduces misalignment across independently predicted planes, leading to more coherent multi-plane aggregation. The training objective is defined as:

$$J_{\text{fut}} = \mathbb{E}_{k,i}[\mathcal{L}_{\text{fut}}(\hat{s}_i^k, s_i^k) + \xi \mathcal{L}_{\text{occ}}(\hat{S}^k, S^k)], \tag{7}$$

where $\mathcal{L}_{\text{fut}} = \mathcal{L}_1 + \mathcal{L}_2$ is the weighted sum of L1 and L2 losses, with $\xi$ as a trade-off weight.

### 3.3 MOTION PLANNING WITH DTT

Given the predicted triplane changes $\Delta\hat{s}^k = \{\Delta\hat{s}_i^k \mid i \in \{xy, xz, yz\}\}$, which encode both global context and local motion dynamics, we directly use them as sparse queries to attentively generate planning outputs across past and future frames, as illustrated in Figure 5.

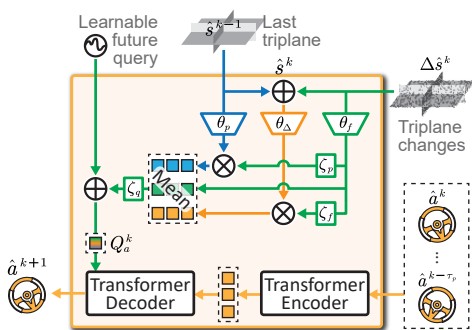

Specifically, the previous triplane $\hat{s}^{k-1}$, the change $\Delta\hat{s}^k$, and the next triplane $\hat{s}^k$ are first mapped into a shared latent space by three ResNet-18 networks (He et al., 2016), $\theta_p$, $\theta_\Delta$, and $\theta_f$, yielding $z^{k-1}$, $\Delta z^k$, and $z^k$. The change feature $\Delta z^k$ is then processed by two parallel fully connected (FC) layers with Sigmoid activations, denoted $\zeta_p$ and $\zeta_f$, to produce query vectors $f^{k-1}$ and $f^k$. These queries are multiplied with $z^{k-1}$ and $z^k$ to extract motion-related features relative to the previous and current triplanes. The resulting features are averaged element-wise with $\Delta z^k$, passed through another FC layer $\zeta_q$, and combined with the learnable future query to form the motion-planning query $Q_a^k \in \mathbb{R}^{d_{\text{act}}}$.

Figure 5: Motion planning with DTT.

Next, we project the actions from the past $\tau_p$ frames into the same dimension as $Q_a^k$, and employ a Transformer encoder to capture motion dependencies. The query $Q_a^k$ is then fed into a Transformer decoder via cross-attention to predict the next action $\hat{a}^{k+1}$:

$$\hat{a}^{k+1} = \mathbf{\Phi}_{\text{act}}(\hat{a}^{k-\tau_p:k}, \hat{s}^{k-1}, \Delta\hat{s}^k, Q_a^k). \tag{8}$$

Unlike OccWorld (Zheng et al., 2024a), which introduces an additional ego token to track the agent's trajectory, our approach learns future motion directly from scene changes and historical motion. This design simplifies the planning module while still delivering safer and more precise motion decisions. The optimization objective is defined as follows:

$$J_{\text{act}} = \mathbb{E}_k[\mathcal{L}_{\text{act}}(\hat{a}^k, a^k)], \tag{9}$$

where $\mathcal{L}_{\text{act}}$ measures the L2 discrepancy between the predicted and GT trajectories.

## 4 EXPERIMENTS

### 4.1 EXPERIMENTAL SETTINGS

**Targets and evaluation metrics.** OWMs aim to jointly modeling occupancy forecasting and motion planning. Following (Zheng et al., 2024a; Wei et al., 2024), we use the past four frames (2 seconds) to predict the outcomes of the next six frames (3 seconds). We conduct two sets of experiments: (i) To evaluate 4D occupancy forecasting, we report intersection over union (IoU) for occupied and unoccupied voxels and mean IoU (mIoU) across 18 semantic classes, based on the occupancy annotations in the Occ3D dataset (Tian et al., 2023). (ii) To assess planning precision and safety, we measure the L2 distance between predicted and GT trajectories (in meters) and the collision rate with traffic participants' bounding boxes, using nuScenes annotations (Caesar et al., 2020).

**Implementation details.** The dataset consists of 1,000 scenes, of which 700 are used for training and 100 for testing. Each scene contains up to 40 timesteps, with a sampling frequency of 2Hz. The occupancy data $S^t$ at each timestep has dimensions of $16 \times 200 \times 200$, while the pre-trained triplane

Table 1: Testing performance comparison with SOTA methods on the 4D occupancy forecasting task. Best values in each metric are **bolded**. 0s refers to reconstruction accuracy, while 1s, 2s, and 3s denote future prediction accuracy. Avg. is the average of 1s, 2s, and 3s.

| Models | Input | mIoU (%) ↑ | | | | | IoU (%) ↑ | | | | |
|---|---|---|---|---|---|---|---|---|---|---|---|
| | | 0s | 1s | 2s | 3s | Avg. | 0s | 1s | 2s | 3s | Avg. |
| OccWorld-O | 3D-Occ | 66.38 | 25.78 | 15.14 | 10.51 | 17.14 | 62.29 | 34.63 | 25.07 | 20.18 | 26.63 |
| OccLLaMA-O | 3D-Occ | 75.20 | 25.05 | 19.49 | 15.26 | 19.93 | 63.76 | 34.56 | 28.53 | 24.41 | 29.17 |
| RenderWorld-O | 3D-Occ | - | 28.69 | 18.89 | 14.83 | 20.80 | - | 37.74 | 28.41 | 24.08 | 30.08 |
| OccLLM-O | 3D-Occ | - | 24.02 | 21.65 | 17.29 | 20.99 | - | 36.65 | 32.14 | 28.77 | 32.52 |
| DOME-O | 3D-Occ | 83.08 | 35.11 | 25.89 | 20.29 | 27.10 | 77.25 | 43.99 | 35.36 | 29.74 | 36.36 |
| DTT-O (ours) | 3D-Occ | **85.50** | **37.69** | **29.77** | **25.10** | **30.85** | **92.07** | **76.60** | **74.44** | **72.71** | **74.58** |
| OccWorld-F | Camera | 20.09 | 8.03 | 6.91 | 3.54 | 6.16 | 35.61 | 23.62 | 18.13 | 15.22 | 18.99 |
| OccLLaMA-F | Camera | 37.38 | 10.34 | 8.66 | 6.98 | 8.66 | 38.92 | 25.81 | 23.19 | 19.97 | 22.99 |
| RenderWorld-F | Camera | - | 2.83 | 2.55 | 2.37 | 2.58 | - | 14.61 | 13.61 | 12.98 | 13.73 |
| OccLLM-F | Camera | - | 11.28 | 10.21 | 9.13 | 10.21 | - | 27.11 | 24.07 | 20.19 | 23.79 |
| DOME-F | Camera | 75.00 | 24.12 | 17.41 | 13.24 | 18.25 | 74.31 | 35.18 | 27.90 | 23.44 | 28.84 |
| DTT-F (ours) | Camera | 43.52 | 24.87 | 18.30 | 15.63 | 19.60 | 54.31 | 38.98 | 37.45 | 31.89 | 36.11 |

shape is $16 \times 100 \times 100$ with 8 channels. In $\Phi_{\text{fut}}$, predictions for each plane use features from $V = 5$ scales, and the token dimension in $\Phi_{\text{act}}$ is $d_{\text{act}} = 50$. The objectives, $J_{\text{enc,dec}}$, $J_{\text{fut}}$, and $J_{\text{mot}}$, are optimized using AdamW, with a weight regularization factor of 0.01, an initial learning rate of 0.001, and cosine decay with a minimum learning rate of $10^{-6}$. We first pre-train $\Phi_{\text{enc}}$ and $\Phi_{\text{dec}}$ with a batch size of 10, using random flip augmentation to obtain the triplane representations. Then, we train $\Phi_{\text{fut}}$ and $\Phi_{\text{act}}$ with a batch size of 1, while fine-tuning $\Phi_{\text{dec}}$. All training and testing are performed on 4 RTX 4090 GPUs. More details are given in the supplementary material.

## 4.2 COMPARISONS WITH THE STATE-OF-THE-ART

**4D occupancy forecasting.** Table 1 reports the testing performance of various methods, under two settings: (i) using 3D occupancy GTs as historical input, marked with "-O"; (ii) using predicted 3D occupancy data from FB-Occ (Li et al., 2023b) as historical input, marked with "-F".

At 0s, baseline methods clearly fall short of ours, mainly because our triplane representation achieves higher-fidelity compression. The only exception is DOME-F, which performs better than DTT-F at this initial step. This is mainly due to the additional noise introduced by the FB-Occ outputs used in our pipeline, while DOME's VAE-style encoder offers better generalization than our triplane encoder. Beyond 0s, both DTT-O and DTT-F leverage triplane-delta predictions instead of full-state predictions, yielding the best mIoU and IoU accuracy with reduced error accumulation. The IoU improvement is particularly significant, as occupancy changes are inherently sparse and predicting binary occupancy (occupied vs. empty) is simpler than multi-class predictions measured by mIoU. Additionally, DTT' strategy of predicting triplane changes significantly aids $\Phi_{\text{fut}}$ in forecasting the next 3 seconds, effectively reducing error accumulation.

**Motion planning.** We compare DTT extensively with SOTA methods for autonomous driving, including LiDAR-based (IL (Ratliff et al., 2006), NMP (Zeng et al., 2019), FF (Hu et al., 2021), and EO (Khurana et al., 2022)), camera-based (ST-P3 (Hu et al., 2022), UniAD (Hu et al., 2023b), VAD (Jiang et al., 2023a)), and occupancy-based methods (OccWorld (Zheng et al., 2024a), RenderWorld (Yan et al., 2025), and OccLLaMA (Wei et al., 2024)). The results on the testing dataset are shown in Table 2. Specifically, (i) LiDAR- and camera-based methods often require additional auxiliary supervisions (e.g., 3D bounding boxes, drivable free space, HD maps) to improve planning quality. (ii) When using camera input only, DTT-F operates as a purely vision-based 4D occupancy forecasting method, with performance dependent on the accuracy of vision-based predictions. In this setting, DTT-F achieves competitive results compared to UniAD. (iii) Occupancy-based methods rely solely on dense 3D occupancy annotations, suggesting that improving the quality of occupancy GT could further enhance planning performance. Compared to SOTA occupancy-based methods, DTT-O delivers superior results, mainly due to its ability to establish precise, task-relevant correlations between scene changes and motion trajectories in a deep latent space, effectively filtering out noise from irrelevant traffic elements. However, for short-term prediction (1–2 seconds), DTT-O underperforms OccLLaMA-O. This is primarily because even slight inaccuracies in the predicted

Table 2: Testing performance of motion planning compared with SOTA method. Best and second-best values in each metric are **bolded** and underlined, respectively. Auxiliary supervision refers to additional supervision signals beyond the GT trajectories.

| Models | Input | Auxiliary supervision | L2 (m) ↓ | | | | Collision rate (%) ↓ | | | |
|---|---|---|---|---|---|---|---|---|---|---|
| | | | 1s | 2s | 3s | Avg. | 1s | 2s | 3s | Avg. |
| IL | LiDAR | None | 0.44 | 1.15 | 2.47 | 1.35 | 0.08 | 0.27 | 1.95 | 0.77 |
| NMP | LiDAR | Box+Motion | 0.53 | 1.25 | 2.67 | 1.48 | 0.04 | 0.12 | 0.87 | 0.34 |
| FF | LiDAR | Freespace | 0.55 | 1.20 | 2.54 | 1.43 | 0.06 | 0.17 | 1.07 | 0.43 |
| EO | LiDAR | Freespace | 0.67 | 1.36 | 2.78 | 1.60 | 0.04 | **0.09** | 0.88 | 0.33 |
| ST-P3 | Camera | Map+Box+Depth | 1.33 | 2.11 | 2.90 | 2.11 | 0.23 | 0.62 | 1.27 | 0.71 |
| UniAD | Camera | Map+Box+Motion+Track+Occ | 0.48 | 0.96 | **1.65** | 1.03 | 0.05 | 0.17 | 0.71 | 0.31 |
| VAD | Camera | Map+Box+Motion | 0.54 | 1.15 | 1.98 | 1.22 | 0.04 | 0.39 | 1.17 | 0.53 |
| OccWorld-F | Camera | None | 0.67 | 1.69 | 3.13 | 1.83 | 0.19 | 1.28 | 4.59 | 2.02 |
| RenderWorld-F | Camera | None | 0.48 | 1.30 | 2.67 | 1.48 | 0.14 | 0.55 | 2.23 | 0.97 |
| OccLLaMA-F | Camera | None | 0.38 | 1.07 | 2.15 | 1.20 | 0.06 | 0.39 | 1.65 | 0.70 |
| DTT-F (ours) | Camera | None | 0.35 | 1.01 | 1.89 | 1.08 | 0.08 | 0.33 | 0.91 | 0.44 |
| OccWorld-O | 3D-Occ | None | 0.43 | 1.08 | 1.99 | 1.17 | 0.07 | 0.38 | 1.35 | 0.60 |
| RenderWorld-O | 3D-Occ | None | 0.35 | **0.91** | 1.84 | 1.03 | 0.05 | 0.40 | 1.39 | 0.61 |
| OccLLaMA-O | 3D-Occ | None | 0.37 | 1.02 | 2.03 | 1.14 | **0.04** | 0.24 | 1.20 | 0.49 |
| DTT-O (ours) | 3D-Occ | None | **0.32** | **0.91** | 1.76 | **1.00** | 0.08 | 0.32 | **0.51** | **0.30** |

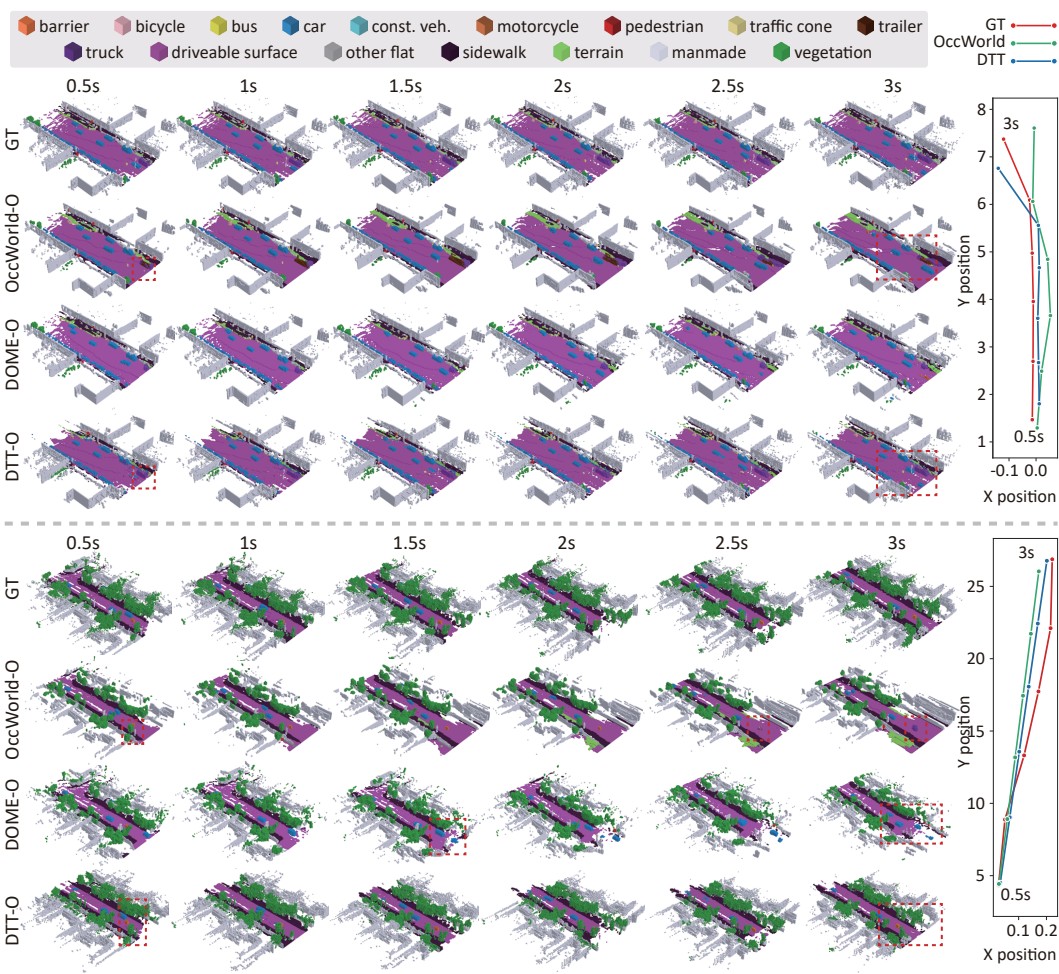

Figure 6: Visualization of 4D occupancy forecasting and motion planning for the next 3 seconds on the Occ3D test dataset (zoom in for a clearer view).

deltas can have a larger impact on near-future action decisions. For instance, a small prediction bias may delay an obstacle-avoidance maneuver, leading to an increased collision rate.

**Visual comparisons with SOTA.** Figure 6 shows the scene evolution over the next 3 seconds with motion predictions for two representative scenarios. Since DOME only performs conditional scene generation, we show its occupancy predictions only. (i) In the first case, OccWorld-O incorrectly labels part of the truck as a trailer, and this error accumulates over time. DOME-O avoids this misclassification through its iterative diffusion-based refinement, but it tends to predict more aggressive motion for the middle vehicles, causing them to leave the frame prematurely. In contrast, DTT-O tracks all objects accurately and preserves clear boundaries throughout the sequence. Nonetheless, our method exhibits some residual artifacts in the 2.0–2.5 s frames for the middle cars. This occurs because the triplane changes preserve fine-grained 3D details, and adding these residuals to the previous triplane can introduce slight decoding noise. By comparison, DOME-O and OccWorld-O rely on VAE-style encoders that produce smoother representations but are less effective at capturing precise object motion. (ii) In the second case, OccWorld-O gradually loses the road boundary and misidentifies the motorcycle as a truck at 2.5 s and 3 s, due to the limited 3D detail preserved by its VAE-based encoder. DOME-O alleviates these issues and better preserves global geometry. However, because it conditions future scene generation primarily on the ego vehicle's trajectory, it may be less sensitive to the detailed motion patterns of other agents, occasionally resulting in drifts or hallucinated objects. In contrast, our method preserves road geometry, roadside structures, and small-object semantics. These precise scene forecasts arise from accurately predicted latent triplane changes, enabling more accurate motion planning.

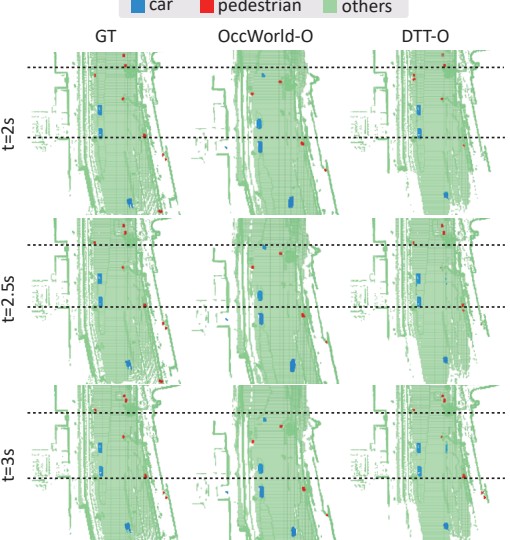

Figure 7: Predicted locations of dynamic objects for the three last frames.

Table 3: Latent representation comparison.

| Type | Latent space ↓ $h, w, l, c$ | Total shape ↓ | mIoU (%) ↑ | IoU (%) ↑ |
|---|---|---|---|---|
| BEV | -, 50, 50, 128 | 320,000 | 60.50 | 59.07 |
| BEV | -, 50, 50, 16 | 40,000 | 37.81 | 46.53 |
| BEV | -, 50, 50, 8 | **20,000** | 35.26 | 42.91 |
| Triplane | 16, 50, 50, 8 | 32,800 | **72.45** | **80.83** |
| BEV | -, 100, 100, 128 | 1,280,000 | 78.12 | 71.63 |
| BEV | -, 100, 100, 16 | 160,000 | 57.34 | 48.19 |
| BEV | -, 100, 100, 8 | **80,000** | 54.13 | 46.78 |
| Triplane | 16, 100, 100, 8 | 105,600 | **85.50** | **92.07** |

Table 4: Ablation study of DTT.

| Idx. | Models | Avg. mIoU ↑ | Avg. L2 ↓ | FPS ↑ |
|---|---|---|---|---|
| **M0** | DTT-O | **30.85** | **1.00** | 26 |
| **M1** | w/o pretraining | 28.45 | 1.12 | 26 |
| **M2** | w/o triplane | 26.71 | 1.13 | 21 |
| **M3** | w/o triplane changes | 27.97 | 1.10 | 27 |
| **M4** | w/o multi-scale mot. | 29.05 | 1.08 | **36** |
| **M5** | w/o autoregression | 29.41 | 1.11 | 34 |

Figure 7 compares occupancy predictions for the three furthest frames, focusing on dynamic objects of varying sizes (cars and pedestrians) from a BEV perspective. Two reference lines are displayed to reflect the errors between the predictions and GT. The results of our method match the GT significantly better than OccWorld-O, which exhibits substantial drift. For instance, the predictions for the pedestrians at the top and the vehicle on the left clearly demonstrate this improvement.

Figure 8 shows two failure cases over three consecutive frames, where fragmented or incomplete road boundaries lead to some deviations in motion planning compared with GT. This mainly stems from the occupancy labels: LiDAR points near road edges are radial and sparse, producing thin, discontinuous boundary surfaces. As the triplane encoder must model both dense interior regions and these sparse edge structures, it tends to prioritize dense areas and underfit boundaries. This limitation could be alleviated by boundary densification or edge-focused regularization during training.

## 4.3 ABLATION STUDY

**Latent representation comparison.** Table 3 empirically finds that adding the two extra planes ($xy$ and $yz$) in BEV allows for fewer channels per plane while still achieving high-quality reconstruction

at 0s. For BEV, the latent size is $w \times l \times c$; for triplane, it is the sum across three orthogonal planes: $(h \times w + h \times l + w \times l) \times c$. These results show that BEV requires significantly more channels to maintain reconstruction quality, while triplane achieves strong performance with fewer channels.

**Effect of DTT components.** Table 4 evaluates the effectiveness of key designs in DTT. **M0** denotes the full model, while ablations are obtained by replacing its components as follows: **M1** removes pre-training and learns triplane changes in an end-to-end manner, causing downstream task gradients to bias the encoder toward task-specific cues while losing 3D scene information, which degrades both encoder quality and overall performance. **M2** uses latent BEV features from OccWorld as input, leading to geometric information loss. **M3** predicts full future triplanes instead of $\Delta$ changes, showing that change prediction is more efficient. **M4** applies a single-scale Transformer, failing to capture multi-scale motion and accumulating errors. **M5** removes the autoregressive mechanism, where each previous prediction is fed into the next step. Instead, it predicts all future outcomes simultaneously from four historical occupancy frames. This prevents step-wise adjustments based on previous predictions, which can degrade performance.

**Effect of long-duration predictions.** In Table 5, we retrain OccWorld, DOME, and DTT for a 10-second prediction task, which is particularly challenging due to the substantial changes that occur in the scene over long horizons. New objects may appear, existing objects can move or leave the field of view, and the overall geometry evolves over time. As a result, the mIoU and L2 metrics of all methods gradually decline with increasing prediction length. In contrast, our approach accumulates errors more slowly, primarily because its multi-scale modeling of triplane deltas allows it to capture fine-grained changes in the scene more effectively across extended durations.

**Effect of different hyperparameters.** Table 6 shows the impact of different hyperparameters, with ▨ indicating our final trade-off. The **H1** series varies the latent triplane shape, defined by

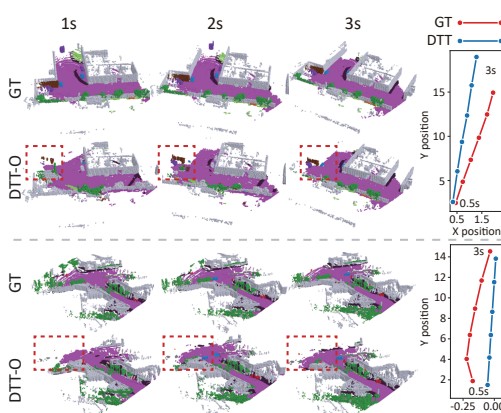

Figure 8: Some failure cases.

Table 5: Effect of long-duration predictions.

|  |  | OccWorld-O | DOME-O | DTT-O |
|---|---|---|---|---|
| mIoU (%) ↑ | 1s | 19.95 | 25.37 | **27.61** |
|  | 3s | 10.65 | 14.13 | **16.22** |
|  | 5s | 6.44 | 9.52 | **11.03** |
|  | 10s | 5.26 | 8.99 | **9.17** |
| L2 (m) ↓ | 1s | 0.88 |  | **0.64** |
|  | 3s | 4.41 | N/A | **3.26** |
|  | 5s | 5.53 |  | **4.43** |
|  | 10s | 5.61 |  | **4.59** |

Table 6: Effect of different hyperparameters.

| Idx. | Setting | Avg. mIoU ↑ | Avg. L2 ↓ | FPS ↑ |
|---|---|---|---|---|
| **H1.1** | 8, (8, 50, 50) | 23.18 | 1.33 | **37** |
| **H1.2** | 8, (16, 50, 50) | 24.56 | 1.23 | 33 |
| **H1.3** | 8, (8, 100, 100) | 27.71 | 1.17 | 30 |
| **H1.4** | 8, (16, 100, 100) | 30.85 | **1.00** | 26 |
| **H1.5** | 16, (16, 100, 100) | **31.02** | 1.09 | 23 |
| **H1.6** | 32, (16, 100, 100) | 29.89 | 1.11 | 18 |
| **H2.1** | $V = 3, 4, 16$ | 29.01 | 1.08 | **28** |
| **H2.2** | $V = 5, 4, 16$ | 30.85 | 1.00 | 26 |
| **H2.3** | $V = 5, 4, 32$ | 31.59 | 1.04 | 21 |
| **H2.4** | $V = 5, 8, 16$ | **32.04** | 1.02 | 23 |
| **H2.5** | $V = 7, 4, 16$ | 28.75 | **0.98** | 20 |

channel number and spatial size $(h, w, l)$; increasing spatial size improves predictions, while more channels offer little benefit and can hurt performance due to feature redundancy in self-attention. The **H2** series adjusts the number of scales $(V)$ and the depth and width of the Transformer, i.e., the number of layers and attention heads. The results show that even a lightweight Transformer captures essential patterns, whereas larger models add cost without notable gains. Our final design uses only 903 M memory (vs. 13,000 M / 13,500 M for OccWorld / RenderWorld) and runs at 26 FPS on an RTX 4090, outperforming OccWorld's 18 FPS.

## 5 CONCLUSION

This paper introduces DTT, a new 4D OWM that leverages pre-trained triplane latent representations and predicts plane-wise future changes with multi-scale Transformers. These predictions are recovered into future occupancy outcomes and used as sparse queries for motion planning. DTT achieve SOTA performance in both scene forecasting and motion planning with real-time efficiency.

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
