# OpenReview forum: "Delta-Triplane Transformers as Occupancy World Models"
_ICLR.cc/2026/Conference — Submitted to ICLR 2026_

### Official Review · Reviewer_foEZ · 2025-10-25

**Soundness:** 2
**Presentation:** 3
**Contribution:** 2
**Rating:** 4
**Confidence:** 3

**Summary:**

This paper introduces Delta-Triplane Transformers (DTT), a new 4D occupancy world model (OWM) for autonomous driving. Unlike previous works (e.g., DOME, OccWorld), DTT does not predict the full occupancy state but instead models changes (deltas) in a compact triplane representation (xy/xz/yz). By leveraging separate multi-scale Transformers per plane, DTT predicts sparse occupancy changes and fuses them to reconstruct future scenes and ego trajectories. The method achieves state-of-the-art results in motion planning and 3D-Occ-based  4D occupancy prediction and runs in real time (26 FPS).

**Strengths:**

1. Triplane representation preserves vertical 3D information and yields a compact latent space, helping reduce drift in long-term prediction.
2. Modeling occupancy deltas is intuitively efficient and effective. As we do not need to "copy" the existing states into the prediction.
3. State-of-the-art results: consistent improvement over DOME, OccWorld, and OccLLaMA, both in accuracy (mainly in 3D-Occ based 4D occupancy prediction) and efficiency.
4. The experiments and supplementary materials are rich. The writing and drawings overall is clear.

**Weaknesses:**

1. Lack of analysis about why learning the "changes" is easier. Firstly, sparse doesn't equals easier (line 085). Then, the full state isn't just hard to learn, but for works like DOME, their error accumulation may additionally comes from the exposure bias. It's intuitive but if we compare the results in Table 1 and Table 3's w/o triplane changes, we can see the mIoU is inferior than DOME. So maybe the slower error accumulation achieved in Table 1, compared with DOME, mainly comes from the triplane representation, rather than the delta estimation?
2. In line 455-456. the author claimed that predicting everything in parallel hinders autoregressive error correction. It's my first time to hear "correction" rather than "accumulation". More discussion about this is welcomed.
3. Lack of analysis about Table 1's camera-based 4D occupancy forecasting. Why DTT is clealy inferior than DOME?

**Questions:**

1. Why compare only the OccWorld in your qualitative experiments and in your supplementary materials? The reviewer considers DOME is a better choice, as its the current state-of-the-art.

---

> ### Author Response · Authors · 2025-11-28
> **Response to Reviewer #foEZ**
>
> We sincerely appreciate your detailed suggestions and constructive comments. We address your concerns point by point as follows:
>
> ---
>
> **W1**: Thank you for the thoughtful and detailed comments. We clarify our viewpoint as follows:
>
> 1. The comparison the reviewer mentioned -- ''Table 1 and Table 3’s w/o triplane changes'' -- seems to refer to DOME-O's Avg. mIoU in Table 1 (27.10) versus M3 in Table 4 (27.97). However, this comparison is not directly meaningful because DOME and our model use different encoders and different prediction architectures. Thus, the performance gap cannot be attributed to whether the model predicts full states or changes.
>
> 2. The benefit of learning changes can be evaluated when the encoder and prediction network are held fixed. This corresponds to the comparison between M0 and M3 in Table 4, where delta prediction yields better accuracy, validating the advantage of modeling scene changes. Moreover, as shown in Fig. 1c, ''changes'' exhibit lower diversity than ''full''' states, making them inherently easier to model because the amount of information to fit is smaller. This is also reflected in the training loss: when predicting full states versus changes, the average triplane loss in Eq. (7) on the training set is:
> |Models|Avg. L_{fut} ↓|
> |--|--|
> |Ours (full prediction)|0.75440|
> |Ours (delta prediction)|0.49532|
>
>
> 3. We further note that when OccWorld is modified to predict changes rather than full future features (as discussed in #WBmZ), its performance also improves. However, the gain is smaller than in our DTT:
>
> |Models|Avg. mIoU (%) ↑|Avg. L2 (m) ↓|
> |--|--|--|
> |Ours (full prediction)|27.97|1.10|
> |Ours (delta prediction)|30.85|1.00|
> |OccWorld (full prediction)|17.14|1.17|
> |OccWorld (delta prediction)|19.02|1.11|
>
> ---
>
> **W2**: Thanks for your suggestion. We have expanded the explanation of M5 in Table 4 and removed the ambiguous term ''correction.'' The updated description can be found in lines 503–508. Specifically, M5 removes the autoregressive prediction mode, where each previous prediction is fed into the next step. Instead, it predicts all future outcomes simultaneously based on the four historical occupancy frames. Consequently, the model loses the ability to make fine-grained adjustments using previous predictions, which can lead to degraded performance.
>
> ---
>
> **W3**: Thank you for your thoughtful comments. Our triplane-based encoder ensures high-fidelity occupancy compression and reconstruction under general conditions. However, when dealing with low-quality or noisy occupancy predictions from camera-based 3D occupancy networks, its robustness is limited, which exposes certain weaknesses of DTT-F. In contrast, the DOME method adopts a VAE-style encoder that offers better robustness to low-quality occupancy inputs. We have added these explanations in the analysis of Table 1, specifically in lines 355–358.
>
> Besides, we observe that when the quality of the occupancy input improves -- for example, when using TPVFormer [1] instead of FB-Occ -- the performance gain of our method is noticeably larger than that of DOME, as shown in the table below. This suggests that our approach will continue to improve as the underlying occupancy predictions become more accurate. We have added this discussion to the appendix.
>
> |mIoU (%) ↑|0s|1s|2s|3s|Avg.|
> |--|--|--|--|--|--|
> |FB-Occ+Ours|43.52|24.87|18.30|15.63|19.60|
> |TPVFormer+Ours|64.13|29.61|20.78|18.41|22.60|
> |FB-Occ+DOME|75.00|24.12|17.41|13.24|18.25|
> |TPVFormer+DOME|76.14|26.58|18.54|13.98|19.70|
>
> [1] Tri-Perspective View for Vision-Based 3D Semantic Occupancy Prediction. CVPR, 2023.
>
> ---
>
> **Q1**: Thank you for your suggestion. When we initially reviewed the DOME paper, we did not find a link to the official implementation and therefore did not include it in our comparison. After re-checking carefully, we realized that the code link is provided in the arXiv comments -- this was our oversight. We have now reproduced DOME and added the comparison in Figure 6 of the main paper, along with the corresponding analysis in lines 432–447. Visual comparisons with DOME are also included in the supplementary material.
>
> The latest state-of-the-art method, DOME, is a strong baseline and provides several insightful design ideas, such as diffusion-based refinement and trajectory resampling, that achieve impressive long-duration performance. Its design provides valuable guidance for future extensions and potential integration with our triplane-based methods. We hope to dive deeper with more discussion with DOME in the future.

---

### Official Review · Reviewer_WBmZ · 2025-10-31

**Soundness:** 2
**Presentation:** 3
**Contribution:** 2
**Rating:** 6
**Confidence:** 5

**Summary:**

This paper introduces Delta-Triplane Transformers (DTT), a novel occupancy world model designed for autonomous driving. The core idea is to represent the 3D environment using triplane latent features and to model occupancy changes (deltas) over time rather than full occupancy states. By leveraging triplane representations, the method preserves vertical (z-axis) structural information that conventional BEV-based approaches tend to lose, avoiding the need for fine-grained semantic cues or large model capacity. Moreover, instead of predicting the entire occupancy state, DTT focuses on modeling occupancy changes (deltas). Since these deltas are sparser and more concentrated around zero, they exhibit lower variance and simplify the learning process. Experiments on the Occ3D and nuScenes datasets demonstrate that DTT achieves SOTA results in both occupancy forecasting and motion planning tasks.

**Strengths:**

1.Comprehensive methodology and clear architecture design.
The paper provides a detailed explanation of the encoder, decoder, delta-based occupancy predictor, and motion planner, as well as how these modules interact within the overall framework. The description of the temporal triplane prediction module, in particular, is well-articulated and technically sound.

2.Novel representation design.
The adoption of triplane as an intermediate latent representation is novel and effective. It addresses key limitations of existing BEV-based occupancy world models, which often lose vertical geometric information and rely on large feature maps for compensation.

3.Impressive results with lightweight architecture.
DTT achieves impressive performance on both Occ3D and nuScenes benchmarks, outperforming current occupancy-based methods while maintaining a smaller model size. This demonstrates strong potential for real-time and practical deployment.

**Weaknesses:**

1.Limited novelty in delta prediction.
Predicting changes (deltas) over time is not a new concept in temporal forecasting. The paper should clarify what makes delta prediction within the triplane representation particularly advantageous compared to delta prediction in other representations.
A suggested ablation study could compare variants such as OccLLaMA or OccWorld, where these methods also predict deltas instead of full states, to isolate the contribution of the triplane-delta combination.

2.Limited discussion on limitations and robustness.
The paper argues that occupancy deltas are sparse and thus easier to learn. However, this assumption might not hold under adverse conditions such as rain, snow, or dense sensor noise, where occupancy changes are no longer sparse. A brief discussion or empirical evidence regarding DTT’s robustness in such noisy or highly dynamic scenarios would strengthen the paper.

3.Inconsistency in figure section titles.
In Section 4.2, the heading “Visual comparisons with SOTA” is inconsistent with the following subsection title “Visual comparisons of motion prediction.” It would be clearer to use a consistent title such as “Visual comparisons of occupancy prediction.” In addition, the visualizations (e.g., Figure 6) could be enlarged to make the differences between methods more clear.

**Questions:**

See above.

---

> ### Author Response · Authors · 2025-11-28
> **Response to Reviewer #WBmZ**
>
> Thank you for your constructive comments on our submission. We address your concerns as follows:
>
> ---
>
> **W1**: Thank you for your constructive comment. To the best of our knowledge, we are the first to adopt delta prediction within occupancy world models (OWMs). We also introduce unique designs specifically for OWMs:
>
> (i) We perform decoupled prediction on the latent triplane features of occupancy, predicting changes separately for each plane.
>
> (ii) We directly add these changes to the previous triplane features for scene forecasting, and also leverage these changes as sparse queries for motion planning.
>
> When OccWorld is modified to predict BEV changes rather than full future features, its performance also improves. However, the gain is smaller than in our triplane-based setting, as shown below.
>
> |Models|Avg. mIoU (%) ↑|Avg. L2 (m) ↓|
> |--|--|--|
> |Ours (full prediction)|27.97|1.10|
> |Ours (delta prediction)|30.85|1.00|
> |OccWorld (full prediction)|17.14|1.17|
> |OccWorld (delta prediction)|19.02|1.11|
>
> We hope the above analysis, combined with the detailed ablation study (Table 4) and the motivation examples in Figure 1 clarify the intuition and our novelty. Please feel free to let us know if further explanation would be helpful.
>
> ---
>
> **W2**: Thank you for the insightful comment and constructive suggestion.
>
> 1. The point raised in our paper (occupancy deltas are sparse and thus easier to learn) was analyzed from a general, dataset-wide statistical perspective (Fig. 1). When considering specific nuScenes subsets under adverse weather conditions (e.g., rainy, foggy, and overcast), the input occupancy data becomes denser for both baseline and our method. Nonetheless, the predicted occupancy changes remain sparser than the dense input, allowing our approach to maintain strong performance on these subsets, as shown below:
>
> |Models|Avg. mIoU (%) ↑|Avg. L2 (m) ↓|
> |--|--|--|
> |OccWorld-F|4.32|1.26|
> |DOME-F|16.44|N/A|
> |Ours-F|18.21|1.10|
>
> 2. In occupancy world model (OWM) frameworks, vision- or LiDAR-based occupancy prediction and scene forecasting/motion planning are decoupled. Occupancy prediction methods primarily address accuracy under adverse conditions. When the quality of the occupancy input improves -- for example, using TPVFormer [1] instead of FB-Occ -- the performance gain of our method is noticeably larger than that of DOME, as shown in the table below. This suggests that our approach will continue to improve as the underlying occupancy predictions become more accurate, consistent with the discussions in #6HR2 Q3 and #S1ww W2.
>
> |mIoU (%) ↑|0s|1s|2s|3s|Avg.|
> |--|--|--|--|--|--|
> |FB-Occ+Ours|43.52|24.87|18.30|15.63|19.60|
> |TPVFormer+Ours|64.13|29.61|20.78|18.41|22.60|
> |FB-Occ+DOME|75.00|24.12|17.41|13.24|18.25|
> |TPVFormer+DOME|76.14|26.58|18.54|13.98|19.70|
>
> 3. Additionally, in the updated manuscript, we provide failure cases (Fig. 8) along with detailed analysis (lines 476–481) to highlight the limitations of our approach.
>
> 4. Finally, we note that the latest state-of-the-art method, DOME, serves as a strong baseline and contains several valuable design ideas, such as trajectory resampling and a diffusion-based architecture, which contribute significantly to long-duration performance. We plan to explore more beneficial integrations with DOME in the future to further advance the OWM field.
>
> ---
>
> **W3**: Thank you for the suggestion. Since both Figure 6 and Figure 7 include results for occupancy prediction and motion planning, we have merged the two subsections into a single one titled ''Visual comparisons with SOTA.'' We have also enlarged Figure 6 and improved its resolution to ensure that it remains clear after zooming in.

---

### Official Review · Reviewer_6HR2 · 2025-11-01

**Soundness:** 3
**Presentation:** 2
**Contribution:** 2
**Rating:** 4
**Confidence:** 5

**Summary:**

The paper proposes Delta-Triplane Transformers (DTT), a 4D occupancy world model that represents scenes via a compact temporal triplane latent and predicts future states by modeling sparse deltas instead of full occupancy, which are then decoded for occupancy forecasting and used as sparse queries for motion planning. DTT pretrains an autoencoder to obtain a triplane latent, applies plane-specific multi-scale Transformers to predict future deltas autoregressively, and couples these with a planning module that attends to change features and history to output trajectories.

**Strengths:**

1. Modeling occupancy deltas in a triplane latent exploits sparsity, reduces variance, and enables lighter sequence models while preserving vertical structure versus BEV-only latents.
2. The performance is better than previous occupancy world models.

**Weaknesses:**

1. The paper claims improvements in motion planning but neglects much of the related work. It does not situate contributions against recent intention-aware or end-to-end planning approaches such as World4Drive, BEV-Planner, GenAD, and PPAD.
2. NuScenes is a small dataset. Would it be possible to evaluate the approach on a larger dataset, such as Waymo [4], or perform the motion planning experiments on NAVSIM?
3. Qualitative results are limited. More diverse and challenging scenarios and explicit failure-case analyses would make the qualitative story more convincing.
4. In Figure 6’s first example, the 2.0s–2.5s frames for the middle cars appear to exhibit stretching/residual artifacts, and in places, OccWorld looks cleaner.

**Questions:**

1. How does DTT perform under closed-loop evaluation and distribution shift (e.g., NavSim or nuPlan), and does the delta modeling reduce compounding error in closed-loop rollouts relative to full-state predictors?
2. Figure 4 caption contains a “traiplane” typo.
3. Why is the reconstruction performance of the camera input setting in the last row in Table 1 lower?

---

> ### Author Response · Authors · 2025-11-28
> **Response to Reviewer #6HR2 [Part 1/2]**
>
> We gratefully appreciate your detailed and constructive suggestions for our work. We have updated the manuscript and provide our response as follows:
>
> ---
>
> **W1**: Thank you for the constructive comment. We have expanded the related work section, and lines 127–137 of the revised manuscript now provide a detailed review of end-to-end autonomous driving methods. Many of these approaches achieve strong performance by leveraging more supervisions [1], high-definition maps [1,4], or richer intention reasoning[2,3,4]. In contrast, our work relies solely on 3D latent triplane changes as a compact conditioning signal for both scene forecasting and motion planning, focusing on occupancy world model comparisons. In the other word, our contributions do not conflict with these planning methods and could potentially enhance each other.
>
> Based on this insight, we conducted a quick verification experiment: after decoding the triplane changes back into occupancy, we added an additional detection head (DH) -- similar to that used in GenAD -- to predict the future positions of surrounding traffic participants. We observed that this further improves performance, as shown below:
>
> |mIoU (%) ↑|0s|1s|2s|3s|Avg.|
> |-|-|-|-|-|-|
> |Ours|85.50|37.69|29.77|25.10|30.85|
> |Ours+DH|91.92|39.14|32.44|27.71|33.09|
>
> |L2 (m) ↓|1s|2s|3s|Avg.|
> |--|--|--|--|--|
> |Ours|0.32|0.91|1.76|1.00|
> |Ours+DH|0.25|0.51|1.04|0.60|
>
> [1] GenAD: Generative End-to-End Autonomous Driving. ECCV, 2024.
> [2] PPAD: Iterative Interactions of Prediction and Planning for End-to-End Autonomous Driving. ECCV, 2024.
> [3] World4Drive: End-to-End Autonomous Driving via Intention-aware Physical Latent World Model. ICCV, 2025.
> [4] Density-adaptive model based on motif matrix for multi-agent trajectory prediction. CVPR, 2024.
>
> ----
>
> **W2**: Thank you for the helpful suggestion. We examined the existing occupancy world models[1-5], and to the best of our knowledge, all prior works are evaluated on NuScenes. To further address the reviewer's concern, we additionally conducted experiments on Waymo [6]. Due to limited time, we were only able to reproduce the occupancy prediction results of OccWorld and our method. The comparison results are as follows:
>
> |mIoU (%) ↑|0s|1s|2s|3s|Avg.|
> |-|-|-|-|-|-|
> |OccWorld-O|60.46|23.14|13.59|10.40|15.71|
> |Ours|82.01|33.21|24.78|22.93|26.99|
>
> |IoU (%) ↑|0s|1s|2s|3s|Avg.|
> |-|-|-|-|-|-|
> |OccWorld-O|58.17|32.82|22.14|20.11|25.02|
> |Ours|90.80|63.43|60.21|58.65|60.76|
>
> Regarding NAVSIM, it does not provide 3D occupancy annotations, and generating reliable occupancy ground truth remains challenging due to the need for accurate geometry, occlusion reasoning, and multi-view consistency. The existing method [6] for producing occupancy labels typically rely on a semi-automatic process, followed by manual inspection to correct holes caused by occlusions. We believe evaluating occupancy world models on NAVSIM would be a valuable direction for future work.
>
> [1] OccWorld: Learning a 3D Occupancy World Model for Autonomous Driving. ECCV, 2024.
> [2] Occ-LLM: Enhancing Autonomous Driving with Occupancy-Based Large Language Models. ICRA, 2025.
> [3] Renderworld: World Model with Self-Supervised 3D Label. ICRA, 2025.
> [4] OccLLaMA: An Occupancy-Language-Action Generative World Model for Autonomous Driving. arXiv, 2024.
> [5] DOME: Taming Diffusion Model into High-Fidelity Controllable Occupancy World Model. arXiv, 2024.
> [6] Occ3D: A Large-Scale 3D Occupancy Prediction Benchmark for Autonomous Driving. NeurIPS, 2023.
>
> ---
>
> **W3**: Thank you for the suggestion. Due to space limitations, we originally included three additional qualitative examples only in the supplementary material. Since ICLR allows adding one extra page during the rebuttal period, we have now incorporated the failure cases and their corresponding analysis into the main text of the revised manuscript (see Fig. 8). Additional qualitative results remain available in the supplementary material.
>
> ---
>
> **W4**: Thank you for the careful observation. Our method does exhibit some residual artifacts in the 2.0–2.5 s frames for the middle vehicles. This occurs because triplane updates preserve fine-grained 3D details, and adding these residuals to the previous triplane may introduce slight decoding noise. In the revised Fig. 6, we have included the latest DOME method as a strong baseline for visual comparison. Overall, our method achieves better quantitative performance, and in practical deployment these residual artifacts can be further reduced through simple post-processing, such as filtering. DOME offers impressive scene-generation quality, and its DiT-based design also inspires us to consider diffusion-style architectures for potential future improvements.

---

> ### Author Response · Authors · 2025-11-28
> **Response to Reviewer #6HR2 [Part 2/2]**
>
> ---
>
> **Q1**: Thank you for the suggestion. As noted in **W2**, neither NavSim nor nuPlan provides dense occupancy ground truth, making quantitative closed-loop evaluation challenging. Nonetheless, we consider closed-loop evaluation an important direction for future work.
>
> ---
>
> **Q2**: Thank you for your careful review. We have revised accordingly.
>
> ---
>
> **Q3**: Thank you for your thoughtful comments. Our triplane-based encoder ensures high-fidelity occupancy compression and reconstruction under general conditions. However, when dealing with low-quality or noisy occupancy predictions from camera-based 3D occupancy networks, its robustness is limited, which exposes certain weaknesses of DTT-F. In contrast, the DOME method employs a VAE-style encoder combined with a diffusion architecture, providing greater robustness to low-quality occupancy inputs. We have added these explanations in the analysis of Table 1, specifically in lines 355–358.
>
> Besides, we observe that when the quality of the occupancy input improves -- for example, when using TPVFormer [1] instead of FB-Occ -- the performance gain of our method is noticeably larger than that of DOME, as shown in the table below. This suggests that our approach will continue to improve as the underlying occupancy predictions become more accurate. We have added this discussion to the appendix.
>
> |mIoU (%) ↑|0s|1s|2s|3s|Avg.|
> |--|--|--|--|--|--|
> |FB-Occ+Ours|43.52|24.87|18.30|15.63|19.60|
> |TPVFormer+Ours|64.13|29.61|20.78|18.41|22.6|
> |FB-Occ+DOME|75.00|24.12|17.41|13.24|18.25|
> |TPVFormer+DOME|76.14|26.58|18.54|13.98|19.7|
>
> [1] Tri-Perspective View for Vision-Based 3D Semantic Occupancy Prediction. CVPR, 2023.

---

### Official Review · Reviewer_S1ww · 2025-11-01

**Soundness:** 3
**Presentation:** 3
**Contribution:** 2
**Rating:** 6
**Confidence:** 4

**Summary:**

This paper proposes Delta-Triplane Transformers (DTT), a 4D Occupancy World Model (OWM) for autonomous driving, aiming to address three key limitations of existing OWMs: loss of vertical spatial information, long-term prediction error accumulation, and excessive model complexity. The core designs of DTT include: 1) A pretrained triplane autoencoder that compactly preserves spatial information of 3D occupancy; 2) Multi-scale Transformers that predict "triplane deltas" (instead of full occupancy states) to leverage the sparsity of changes for reduced modeling difficulty; 3) A sparse query-based motion planning module designed using these deltas to simplify the decision-making process. Experiments on the nuScenes and Occ3D datasets validate DTT’s superiority: compared to the state-of-the-art (SOTA) method DOME, DTT improves mean IoU (mIoU) from 27.10% to 30.85% and IoU from 36.36% to 74.58, while achieving real-time inference on an RTX 4090 GPU.

**Strengths:**

1. **Clarity of Writing:** The paper is very clearly structured and written, allowing readers to easily follow the authors' reasoning about the DTT method and its components.
2. **Clear Motivation:** The motivation for DTT is well-justified—by adopting incremental modeling and leveraging sparsity (via delta prediction), the method significantly reduces computational burden compared to full-state prediction approaches.
3. **Relevance to Research Needs:** The research addresses a topic of high interest in autonomous driving. The achievement of real-time inference on an RTX 4090 makes DTT more relevant to real-world deployment than slower baselines.
4. **Compelling Experimental Results:** The experiments clearly demonstrate that DTT improves both computational efficiency and core performance metrics, avoiding the common trade-off between speed and accuracy.

**Weaknesses:**

1. **Insufficient Theoretical Analysis of Methodology:** The paper primarily uses experimental results to validate its design choices—for example, pretraining the triplane model first and then fixing the encoder to train DTT and the decoder. While Table 4 shows that omitting pretraining degrades performance, the authors lack an intuitive theoretical analysis to explain why this is the case. For instance, Why does end-to-end training (without separating pretraining and fine-tuning) not yield better results? Or is end-to-end training too challenging (e.g., due to high dimensionality or unstable gradients) and prone to local convergence?
2. **Incomplete Analysis of Cross-Input Performance:** From Table 1, DTT achieves significant performance gains when the input is 3D occupancy ground truth (3D-Occ). However, when the input is replaced with camera-derived data (Camera), the improvement margin shrinks substantially—for example, DOME-F outperforms DTT-F significantly at 0s and 1s. The authors need to provide a detailed explanation for this discrepancy (e.g., whether camera-based 3D occupancy predictions introduce noise that disproportionately affects DTT’s delta-based modeling).
3. **Cumulative Error Risks:** DTT adopts an autoregressive framework that predicts the next frame using historical information. This raises a critical question: Does DTT still suffer from cumulative error over time? If the model is tasked with predicting longer occupancy sequences (e.g., beyond 3 seconds), will severe prediction drift occur? The current experiments only evaluate up to 3 seconds, and no analysis of long-horizon stability is provided.
4. **Motion Planning Performance Trends:** In terms of collision rate (Table 2), DTT-O performs significantly worse than OccLLaMA-O at 1s and 2s but outperforms it at 3s. The authors need to explain this counterintuitive trend. For example, is DTT less effective at short-term (near-future) frame prediction, and if so, what causes this delay in performance improvement?

**Questions:**

Please refer to the "Weaknesses" section for detailed questions and suggestions.

---

> ### Author Response · Authors · 2025-11-28
> **Response to Reviewer #S1ww**
>
> We sincerely appreciate the reviewer's constructive feedback and positive remarks on our work. We provide the following detailed responses to your major concerns.
>
> ----
>
> **W1**: Thank you for the insightful comment. Although our explanation is based on empirical observations, there are two underlying theoretical intuitions:
> 1. In a fully end-to-end setup, the supervision from downstream tasks (including occupancy forecasting and motion planning) tends to push the encoder to extract only task-specific features, causing it to lose much of the rich 3D scene information.
> 2. Pretraining the encoder ensures that the prediction module (DTT) learns future prediction on top of a stable and well-initialized representation. This prevents mutual interference between the encoder and DTT during joint optimization.
>
> We have emphasized these points in the revised Section 3.2.2 (lines 270–273) and in the analysis of Table 4 (lines 494–497). The reviewer may refer to our updated manuscript.
>
> ----
>
> **W2**: Thank you for your thoughtful comments. Our triplane-based encoder ensures high-fidelity occupancy compression and reconstruction under general conditions. However, when dealing with low-quality or noisy occupancy predictions from camera-based 3D occupancy networks, its robustness is limited, which exposes certain weaknesses of DTT-F. In contrast, the DOME method employs a VAE-style encoder combined with a diffusion architecture, providing greater robustness to low-quality occupancy inputs. We have added these explanations in the analysis of Table 1, specifically in lines 355–358.
>
> Besides, we observe that when the quality of the occupancy input improves -- for example, when using TPVFormer [1] instead of FB-Occ -- the performance gain of our method is noticeably larger than that of DOME, as shown in the table below. This suggests that our approach will continue to improve as the underlying occupancy predictions become more accurate. We have added this discussion to the appendix.
>
> |mIoU (%) ↑|0s|1s|2s|3s|Avg.|
> |--|--|--|--|--|--|
> |FB-Occ+Ours|43.52|24.87|18.30|15.63|19.60|
> |TPVFormer+Ours|64.13|29.61|20.78|18.41|22.60|
> |FB-Occ+DOME|75.00|24.12|17.41|13.24|18.25|
> |TPVFormer+DOME|76.14|26.58|18.54|13.98|19.70|
>
> [1] Tri-Perspective View for Vision-Based 3D Semantic Occupancy Prediction. CVPR, 2023.
>
>
> ----
>
> **W3**: Thanks for the suggestion. We originally followed the settings used in most occupancy world model papers [1,2,3,4], which evaluate performance at horizons up to 3 seconds. However, maintaining consistency and stability beyond 3 seconds is inherently challenging [5], as scenes can change drastically with new objects appearing and others leaving. Our original Table 5 reported results at 5 seconds; we have now extended the evaluation to 10 seconds to better illustrate long-horizon error accumulation. The reviewer may refer to our updated manuscript. As shown in the revised Table 5, the performance of all methods degrades over time; however, our approach exhibits a noticeably slower accumulation of errors. This advantage primarily stems from our multi-scale modeling of scene changes, which allows our method to better capture fine-grained changes in the environment.
>
> [1] OccWorld: Learning a 3D Occupancy World Model for Autonomous Driving. ECCV, 2024.
> [2] Occ-LLM: Enhancing Autonomous Driving with Occupancy-Based Large Language Models. ICRA, 2025.
> [3] Renderworld: World Model with Self-Supervised 3D Label. ICRA, 2025.
> [4] OccLLaMA: An Occupancy-Language-Action Generative World Model for Autonomous Driving. arXiv, 2024.
> [5] DOME: Taming Diffusion Model into High-Fidelity Controllable Occupancy World Model. arXiv, 2024.
>
> ----
>
> **W4**: Our DTT module is designed to model scene changes rather than reconstruct the full scene content. As a result, the accumulated prediction error of DTT grows more slowly compared with methods that directly model the complete occupancy (e.g., OccLLaMA-O). This explains why DTT-O achieves the lowest collision rate at the 3-second horizon.
>
> For the slightly worse collision rates at 1s and 2s, the differences are extremely small (0.04% at 1s and 0.08% at 2s). In nuScenes, there are 150 test scenes with up to 40 steps each (6000 steps total), so a 0.08% gap corresponds to only about four mismatched samples. We believe these discrepancies fall within normal fluctuations in model behavior.
>
> To better highlight the early-horizon collision-rate differences, we have added representative failure cases in the revised manuscript (Fig. 8 and the analysis in lines 476–481).

---

### Meta-Review · Area_Chair_VcGk · 2026-01-06

**Summary:**

The reviewers acknowledged DTT's computational efficiency and the use of triplane representations to preserve vertical structure compared to BEV-only methods. However, the primary concerns are on the method's significant performance drop in camera-based settings compared to state-of-the-art models (DOME should be excluded as a contemporaneous arXiv paper), visible visual artifacts (e.g., ghosting) arising from the delta-accumulation mechanism, and a lack of strong theoretical justification for the "delta" learning premise. The paper received a mixed initial score of (6, 4, 6, 4). While the authors provided additional baselines and datasets in the rebuttal, the limitations in robustness and rendering quality may still remain unresolved. Thus, the recommendation is rejection.

**Reviewer Concerns:**

Addressed concerns:
- Missing baselines: the authors added comparisons with DOME and additional experiments on the Waymo dataset, addressing specific requests from reviewers
- Long-term stability: the authors extended evaluations to 10 seconds to demonstrate error accumulation trends, addressing the concern about long-horizon drift.

Concerns might still be outstanding:
- Camera-based performance: the method exhibits a significant performance gap when transitioning from ground-truth occupancy to practical camera-based inputs, suggesting a lack of robustness in the triplane encoder.
- Visual artifacts: the "delta" prediction strategy introduces persistent stretching and artifacts on moving vehicles, which degrade visual fidelity compared to full-state prediction methods.
- Theoretical validity: the authors claim occupancy deltas are sparse and thus easier to learn, but the reviewers raised a concern about this part. The reviewer noted that sparse targets do not necessarily equate to easier optimization and that delta prediction can hinder error correction mechanisms found in other architectures.

**Reviewer Scores:**

The manuscript first received an initial review score of  (6, 4, 6, 4), and none of the reviewers participated in the discussion phase.

The authors tried to address all concerns/questions from reviewers. However, given the large quantity of concerns/questions and those (likely) unresolved concerns/questions, I am not confident that reviewers would raise their scores. And thus, I would assume a final average score of 5-5.5. Considering also concerns that might still be outstanding, I recommend rejection of the paper.

---

### Decision · Program_Chairs · 2026-01-26

Reject